# Stagnation of life expectancy in Korea in 2018: A cause-specific decomposition analysis

**Taejae Kim**[1], **Jinwook Bahk**[2], **Hwa Kyung Lim**[3], **Young-Ho Khang**[1,3]*

**1** Department of Health Policy and Management, Seoul National University College of Medicine, Seoul, South Korea, **2** Department of Public Health, Keimyung University, Daegu, South Korea, **3** Institute of Health Policy and Management, Seoul National University Medical Research Center, Seoul, South Korea

* yhkhang@snu.ac.kr

**Data Availability Statement:** All data used in this study are publicly available from the Korea Statistical Information Service (http://kosis.kr/index/index.do).

## Abstract

According to the most recent annual report released by Korea Statistics, the life expectancy at birth (for both sexes) in 2018 was 82.7 years, an increase of 0.0 years over 2017, reflecting the first stagnation in life expectancy since 1960. In this study, a time-series analysis was conducted of trends in life expectancy from 2003 to 2018, and causes of death were analyzed using the Kannisto-Thatcher method and the Arriaga decomposition method. The time trend analysis of yearly life expectancy changes indicated that, in Korea, there was a tendency for the yearly increase in life expectancy between 2003 and 2018 to decrease by 0.0211 years per calendar year. The contribution of cardiovascular diseases, the most important contributor to the life expectancy increase in Korea, gradually decreased over this period. The contribution of cardiovascular diseases to the life expectancy increase was 0.506 years in 2003–2006, but this contribution decreased to 0.218 years in 2015–2018. The positive contributions of ill-defined causes and external causes to life expectancy increase detected in previous periods were not evident in 2015–2018. Diseases of the respiratory system made the largest negative contribution both between 2015 and 2018 and between 2017–2018. The life expectancy stagnation in 2018 could be understood as the combined effect of (a) decreasing momentum in the increase of life expectancy and (b) a chance event in 2018 involving life expectancy. Currently, it is difficult to judge whether the stagnation of life expectancy in 2018 is temporary, and further analyses of life expectancy and contributing causes of death in the future are needed.

## Introduction

Life expectancy in South Korea (hereafter, 'Korea') increased at an unprecedented rate from 52.4 years in 1960, which was 16 years below the Organization for Economic Co-operation and Development (OECD) average at the time, to 62.2 years in 1970 and 78.5 years in 2005 [1]. Life expectancy in Korea reached 82.7 years in 2017, with an average annual increase of 0.46 years from 1970 to 2017, corresponding to one of the highest rates of increase among OECD member countries, along with Turkey (average annual increase of 0.54 years) and Chile (average annual increase of 0.40 years) [2]. Although the magnitude of the annual increase in life expectancy has varied, no stagnation or decrease in life expectancy has been recorded in Korea in the past decades.

**Funding:** This research was supported by a grant of the Korea Health Technology R&D Project through the Korea Health Industry Development Institute, funded by the Ministry of Health & Welfare, Republic of Korea (grant number: HI18C0446). The funder had no role in study design, data collection and analysis, decision to publish, or preparation of the manuscript.

**Competing interests:** NO. The authors have declared that no competing interests exist.

However, according to the life table from Statistics Korea in 2018, the life expectancy at birth for both sexes in 2018 was 82.7 years, a null increase over the previous year [3]. Statistics Korea found that the risk of mortality from pneumonia contributed to the stagnation in life expectancy and stated in a policy briefing that the cold winter season in 2018 led to increased mortality from pneumonia [4]. This explanation implies that the observed stagnation was the temporary consequence of weather-related factors; however, the stabilization of life expectancy after an extended period with such a high rate of increase is highly significant and warrants a more in-depth analysis of potential causes beyond a surface-level analysis of winter temperatures.

In order to understand the causes of the recent stagnation in life expectancy in Korea, it is necessary to examine which age groups and causes of death contribute the most to this phenomenon. One possible explanation for the stagnation would be that Korea's life expectancy in 2017 is already high (82.7 years) and has reached its limit. Age- and cause-specific contributions to the life expectancy increase may also provide an answer to such explanations. This study conducted a time-series analysis of life expectancy increases and trends from 2003 to 2018 and a decomposition analysis of causes of death by age category from ages 0 to 90 and above to identify the factors that contributed, positively or negatively, to changes in life expectancy.

## Materials and methods

### Data sources

This study was exempted from ethical review by the Seoul National University Hospital Institutional Review Board (IRB No. E-2003-098-1109). Aggregate population and mortality data grouped by calendar year, sex, and age groups (and causes of death for mortality data) were provided by the Korean Statistical Information Service (KOSIS). The data included the registered population at mid-year by location, sex, and age, as well as the mortality rate and numbers by cause of death (236 items), sex, and age. The total population and number of deaths from 2003 to 2018 are shown in S1 Table. Age groups were categorized by 5-year increments, except that newborns (age 0) were categorized separately (0, 1–4, 5–9, . . ., 85–89, older than 90). Population and death data used for the analysis are included in the S1 Appendix.

Previous research on life expectancy has demonstrated that individuals older than 80 contribute the most to life expectancy estimates due to population aging [5], a phenomenon that warrants a more detailed analysis of older age groups in 5-year increments like those used for younger age groups. Data on the total population and mortality by age are available from 1983 through KOSIS, but only since 2000 have mortality data on individuals older than 80 been further stratified into age groups of 80–84, 85–90, and 90 and above. Analyses of mortality data by age are further complicated by deaths among newborn children (0 years of age). Parents are required by law to register newborn children within 1 month of their birth. Therefore, the mortality data of newborn children who die within 1 month of birth may be missing. The birth and death records of children who live beyond the first month have also been found to be frequently missing [6]. Since 2000, the KOSIS mortality data have included information from supplemental data collection on causes of death, enabling a more accurate analysis of neonatal deaths without correction [7]. Meanwhile, the Human Mortality Database indicated a potential under-registration problem for infant mortality data between 2000 and 2002 [8]. This study therefore used mortality data by cause of death, sex, and age from 2003 to 2018, with no corrections of the number of deaths by causes of death. In this paper, the use of the population and death data from KOSIS resulted in slight differences in life expectancy and its annual changes from the official report by Statistics Korea. Causes of death were classified according

to the summary table of the Korea Standard Classification of Causes of Death (17 broad causes and 112 detailed causes) as provided by the KOSIS.

## Data analysis

We constructed annual abridged life tables with 5-year age groups and calculated life expectancy at birth using the life table method. The mortality rate in the age category of 90 years and above was corrected using the Kannisto-Thatcher method before analysis [9]. The Kannisto-Thatcher method has been used in previous Korean studies [5, 10]. The contributions of age group and cause of death to life expectancy at birth were calculated using the Arriaga method [11]. The Arriaga method is a decomposition analysis method that decomposes mortality by age group and by cause of death within each age group. The contribution of each age group to differences in life expectancy at birth can be calculated using the contribution of each cause of death within each age group and in the total population. Previously, we employed the Arriaga method to analyze changes in life expectancy [1] and socioeconomic differences in life expectancy [5]. All analyses were conducted using SAS 9.4 (SAS Institute, Inc., Cary, NC, USA).

## Results

Fig 1 illustrates the trends in life expectancy at birth by sex from 2003 to 2018. The life expectancy in Korea increased at an average yearly rate of 0.374 years (0.404 years for males and 0.338 for females) from 2003 to 2018. From 2003 to 2017, the annual increase in life expectancy ranged between 0.144 years and 0.579 years, and the mean annual rate of increase was 0.398 years (0.426 years for males and 0.361 years for females). However, from 2017 to 2018, the rate of increase slowed to 0.041 years (0.093 for males and 0.020 for females), which was 0.357 years lower (0.333 for males and 0.341 for females) than the mean annual rate of increase from 2003 to 2017.

Fig 2 presents time trends in yearly life expectancy changes in Korea between 2003 and 2018, indicating a decreasing trend in yearly life expectancy changes. The beta coefficient (slope) of yearly changes in life expectancy per calendar year between 2003 and 2018 was -0.0211 and the beta coefficient between 2003 and 2017 was -0.015. In addition, we conducted an outlier test, using the outlierTest () function in the *car* package in R, for life expectancy in 2018 based on the regression analysis of life expectancy by year. This test indicated that the

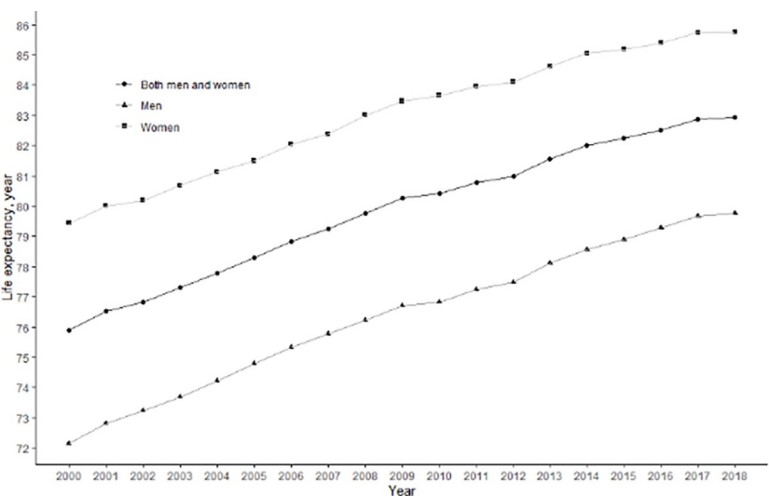

**Fig 1. Trends in life expectancy by sex in Korea between 2003 and 2018.**

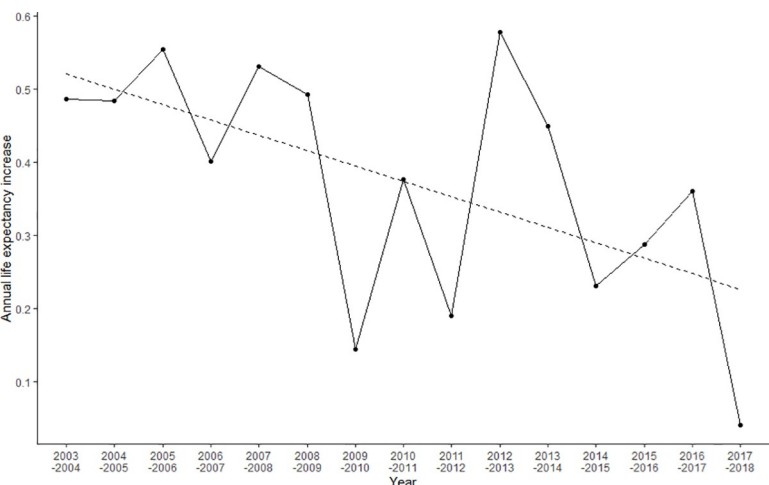

**Fig 2. Time trends in yearly life expectancy changes in Korea between 2003 and 2018.**

studentized residual for the year 2018 was the largest, but the Bonferroni P-value for the null hypothesis (H0: life expectancy in 2018 was not an outlier) was 0.292, implying that the life expectancy in 2018 was not an outlier.

As shown in Table 1, we divided the period of this study into four sub-periods and presented age-specific contributions to the change in life expectancy for 2003–2006, 2007–2010, 2011–2014, and 2015–2018. The magnitude of life expectancy increase was smaller in 2015–2018 (0.690 years) than in the prior three sub-periods. The comparatively smaller increase in life expectancy between 2015 and 2018 can be explained by smaller age-specific contributions from most of the adulthood and elderly age groups (e.g., age groups of 50+) in 2015–2018 than the corresponding age-specific contributions in any of the other three sub-periods. The contribution from the 70–74 and 75–79 age groups in 2015–2018 accounted for 43.5% of the total life expectancy increase.

Fig 3 presents age-specific contributions to the change in life expectancy at birth from 2017 to 2018 and to the average annual change in life expectancy at birth from 2003 to 2017. The decomposition analysis of age-specific contributions to the change in life expectancy from 2017 to 2018 indicated that the 70–74 age group made the largest positive contribution (0.006 years). From 2003 to 2017, the yearly average contribution from the 75–79 age group was also the highest among all age groups (0.052 years). However, with the exception of the 70–74 age group, the age-specific contributions to the change in life expectancy at birth from 2017 to 2018 were lower than the corresponding average yearly age-specific contributions from 2003 to 2017. The age group with the lowest contribution to the increase in life expectancy at birth from 2017 to 2018 was the 90 and above category (-0.009 years), followed by the age groups of 0, 30–34, and 20–24 years. S2 Table shows the results of the broad cause-specific decomposition analysis of the changes in life expectancy in four sub-periods (2003–2006, 2007–2010, 2011–2014, and 2015–2018). The contribution of cardiovascular disease gradually decreased during the four sub-periods, from 0.506 years in 2003–2006 to 0.218 years in 2015–2018. However, the contribution of cancer was relatively stable over the sub-periods, and cancer made the largest positive contribution (0.291 years) between 2015 and 2018. The positive contribution by ill-defined causes was very large in both 2007–2010 (0.325 years) and 2011–2014 (0.275 years), but became negative (-0.016 years) in 2015–2018. Of 17 broad causes of death, diseases of the respiratory system made the largest negative contribution (-0.088 years) between 2015

**Table 1. Age-specific contributions to the change in life expectancy in four sub-periods (2003–2006, 2007–2010, 2011–2014, and 2015–2018) in Korea.**

| Age group | 2003–2006 | | 2007–2010 | | 2011–2014 | | 2015–2018 | |
|---|---|---|---|---|---|---|---|---|
| | Years | % | Years | % | Years | % | Years | % |
| 0 | 0.102 | 6.7 | 0.029 | 2.5 | 0.007 | 0.6 | 0.001 | 0.1 |
| 1–4 | 0.020 | 1.3 | 0.019 | 1.6 | 0.017 | 1.4 | 0.008 | 1.2 |
| 5–9 | 0.013 | 0.9 | 0.013 | 1.1 | 0.004 | 0.3 | 0.005 | 0.7 |
| 10–14 | 0.003 | 0.2 | 0.003 | 0.3 | 0.013 | 1.1 | -0.003 | -0.4 |
| 15–19 | 0.026 | 1.7 | 0.011 | 0.9 | 0.008 | 0.7 | -0.002 | -0.3 |
| 20–24 | 0.031 | 2.0 | 0.009 | 0.8 | 0.026 | 2.1 | -0.001 | -0.1 |
| 25–29 | 0.034 | 2.2 | -0.002 | -0.2 | 0.038 | 3.1 | 0.016 | 2.3 |
| 30–34 | 0.048 | 3.1 | 0.002 | 0.2 | 0.017 | 1.4 | 0.008 | 1.2 |
| 35–39 | 0.060 | 3.9 | -0.002 | -0.2 | 0.015 | 1.2 | 0.001 | 0.1 |
| 40–44 | 0.079 | 5.2 | 0.026 | 2.2 | 0.033 | 2.7 | 0.024 | 3.5 |
| 45–49 | 0.076 | 5.0 | 0.029 | 2.5 | 0.049 | 4.0 | 0.034 | 4.9 |
| 50–54 | 0.094 | 6.2 | 0.034 | 2.9 | 0.050 | 4.1 | 0.040 | 5.8 |
| 55–59 | 0.082 | 5.4 | 0.073 | 6.3 | 0.063 | 5.2 | 0.034 | 4.9 |
| 60–64 | 0.157 | 10.3 | 0.096 | 8.2 | 0.103 | 8.4 | 0.062 | 9.0 |
| 65–69 | 0.156 | 10.2 | 0.137 | 11.7 | 0.145 | 11.9 | 0.062 | 9.0 |
| 70–74 | 0.173 | 11.3 | 0.139 | 11.9 | 0.152 | 12.5 | 0.173 | 25.1 |
| 75–79 | 0.175 | 11.5 | 0.183 | 15.7 | 0.128 | 10.5 | 0.127 | 18.4 |
| 80–84 | 0.107 | 7.0 | 0.186 | 15.9 | 0.121 | 9.9 | 0.056 | 8.1 |
| 85–89 | 0.066 | 4.3 | 0.115 | 9.8 | 0.129 | 10.6 | 0.045 | 6.5 |
| 90+ | 0.023 | 1.5 | 0.070 | 6.0 | 0.100 | 8.2 | 0.001 | 0.1 |
| Total life expectancy increase | 1.525 | 100.0 | 1.168 | 100.0 | 1.219 | 100.0 | 0.690 | 100.0 |

and 2018. The contribution of external causes to the life expectancy increase was relatively large in 2003–2006 (0.253 years) and 2011–2014 (0.267 years), but became small in 2007–2010 (0.020 years) and 2015–2018 (0.095 years).

Fig 4 presents broad cause-specific contributions to the change in life expectancy between 2017 and 2018 and to the average annual change in life expectancy between 2003 and 2017.

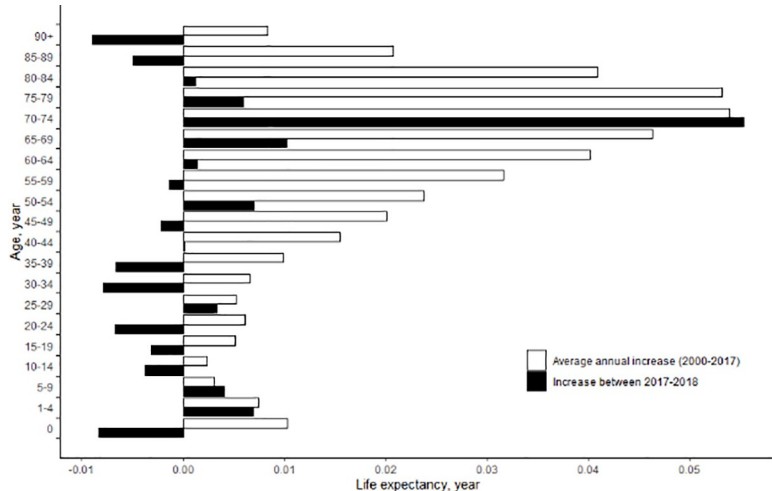

**Fig 3. Age-specific contributions to the change in life expectancy between 2017 and 2018 and to the average annual change in life expectancy between 2003 and 2017 in Korea.**

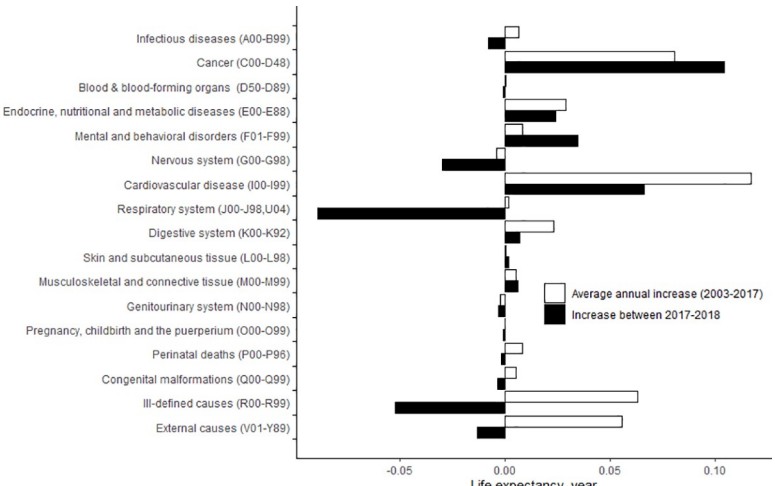

**Fig 4. Broad cause-specific contributions to the change in life expectancy between 2017 and 2018 and to the average annual change in life expectancy between 2003 and 2017 in Korea.**

Negative contributions to life expectancy were made by diseases of the respiratory system (-0.089 years), ill-defined causes (-0.052 years), diseases of the nervous system (G00-G98) (-0.030 years), and external causes (-0.013 years). Compared to the average positive annual contributions between 2003 and 2017, the negative contributions of ill-defined causes and external causes were noteworthy. The magnitude of the positive contribution of cardiovascular diseases decreased between 2017 and 2018. In contrast, the contribution of cancer (0.105 years) between 2017 and 2018 was somewhat greater than its annual contribution between 2003 and 2017.

Fig 5 illustrates the contributions of pneumonia (J12-J18) to the change in life expectancy from 2017 to 2018 and to the average annual change in life expectancy from 2003 to 2017 by age group. The contribution of pneumonia to the change in life expectancy from 2017 to 2018 was -0.078 years, which was 0.057 years lower than the contribution of pneumonia to the

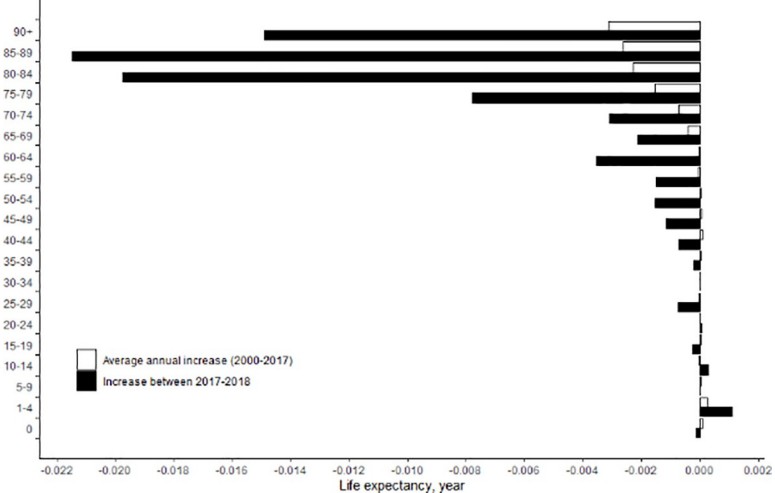

**Fig 5. Contributions of pneumonia to the change in life expectancy between 2017 and 2018 and to the average annual change in life expectancy between 2003–2017 by age group in Korea.**

average annual change in life expectancy from 2003 to 2017 (-0.021 years). This difference accounted for 21.8% of the change (-0.357 years) in the change of life expectancy from 2017 to 2018 from the average annual change in life expectancy from 2003 to 2017. When the contributions of pneumonia to life expectancy were analyzed by age group, from 2003 to 2017, the most negative contribution was made by the 90 and above age group (-0.005 years, 23.9%), whereas from 2017 to 2018, the most negative contribution was from the 85–89 age group (-0.021 years, 27.7%). When we examined the contribution of pneumonia to the change in life expectancy for four sub-periods (S3 Table), consistent negative contributions to the life expectancy increase were noticed, but the negative contribution in 2015–2018 was the greatest among the four sub-periods.

## Discussion

This study examined the contributions of various age groups and causes of death to the stagnation of life expectancy in 2018 using life expectancy decomposition analyses based on population and mortality data from 2003 to 2018 in Korea.

The time trend analysis of yearly life expectancy changes (Fig 2) indicated that, in Korea, the yearly increase in life expectancy between 2003 and 2018 tended to decrease by 0.0211 years per calendar year. The outlier test also revealed that the life expectancy in 2018 was not an outlier, although the studentized residual for the year 2018 was the largest. When we divided the period into four sub-periods (2003–2006, 2007–2010, 2011–2014, and 2015–2018), the magnitude of the life expectancy increase was smaller in 2015–2018 (0.690 years) than in the prior three sub-periods. The stagnation of life expectancy in 2018 could be understood as the combined effect of (a) a long-term decreasing trend in the increase in life expectancy and (b) a chance event in 2018 involving life expectancy.

When age-specific contributions to the change in life expectancy were analyzed in the four sub-periods, it was found that age-specific contributions in most of the adulthood and elderly age groups (e.g., age groups of 50+) were smaller in 2015–2018 than the corresponding age-specific contributions in any of the other three sub-periods. An age-specific decomposition analysis for the period 2017–2018 also showed that the contributions of most age groups (except for the 70–74 age group) to the increase in life expectancy from 2017 to 2018 were lower than their contributions to the average annual increase in life expectancy from 2003 to 2017. These findings imply that the stagnating life expectancy in 2018 might have resulted from a decreasing rate of mortality reduction in causes of death in older age groups.

Our analysis of cause-specific contributions to increases in life expectancy showed that the contribution of cardiovascular diseases (I00-I99), the most important contributor to life expectancy increase in Korea [1], gradually decreased over the period of the study (S2 Table). The contribution of cardiovascular diseases was 0.506 years in 2003–2006, 0.448 years in 2007–2010, 0.349 years in 2011–2014, and 0.218 years in 2015–2018. The finding suggests that the reduction in mortality from cerebrovascular diseases is reaching its limits, partly supporting an explanation that Korea's life expectancy is already high. Cause-specific life expectancy decomposition analyses (S2 Table) also revealed that the positive contributions of ill-defined causes and external causes to the increase in life expectancy detected in previous periods were not evident in 2015–2018.

Diseases of the respiratory system made the largest negative contribution both between 2015 and 2018 (S2 Table) and between 2017–2018 (Fig 4). Of the diseases of the respiratory system, pneumonia was the main contributor (S3 Table). The contribution of pneumonia to the change in life expectancy from 2015 to 2018 was -0.117 years (S3 Table). In addition, the contribution of pneumonia from 2017 to 2018 was -0.077 years, which was significantly lower

than its contribution to the average annual change in life expectancy from 2003 to 2017, which was -0.021 years. In Korea, pneumonia was ranked the 10th most common cause of death, with 3,506 deaths in 2004, but was ranked higher in 2015 at fourth, with 14,718 deaths [12]. Pneumonia can be classified by its route of acquisition into community-acquired pneumonia and hospital-acquired pneumonia. The causative micro-organism of community-acquired pneumonia is *Streptococcus pneumoniae*, and influenza is the most common viral cause [13]. Streptococcal pneumonia is a legally notifiable communicable disease that has steadily become more common, from 228 infections in 2015 to 441 in 2016, 523 in 2017, and 670 in 2018. In 2018, 2,034 cases of influenza were detected through sample monitoring [14], and a recent age-period-cohort analysis showed that, after controlling for age and cohort effects, the period effect gradually increased from 2014 to 2018 [15]. The increasing frequency of community-acquired pneumonia is understood to be driven by the aging population and the increasing number of people living with chronic diseases [16]. The population of Korea is aging rapidly, and institutionalization for medical care is becoming more common, as shown by increases in long-term care hospitals and care facilities. Pneumonia acquired by elderly patients while they are institutionalized at care hospitals and facilities has been researched in observational studies [17, 18].

This study has several limitations. Our decomposition analyses employed aggregate population and mortality data, and estimation of confidence intervals for the decomposition analysis using individual level data was not possible [19]. Another limitation is that because this analysis used the classification of causes of death from Statistics Korea, it was not possible to re-classify ill-defined causes (R00-R99). Despite these limitations, this study is valuable that it brings the issue of the stagnation of life expectancy in 2018 to the attention of researchers and that it provides basic data for future studies.

## Conclusion

In conclusion, the life expectancy stagnation in 2018 could be a result of the combined effect of (a) long-term decreasing momentum in the increase in life expectancy and (b) a chance event in 2018 involving life expectancy. A gradually decreasing trend in the contribution of cardio-vascular diseases to the increase in life expectancy might have significantly contributed to this stagnation. In addition, a more pronounced upward trend in pneumonia deaths in 2018 also was another contributor. Ill-defined causes and external causes, which made positive contributions to the life expectancy increase in previous periods, did not make any positive contributions between 2015 and 2018 and between 2017 and 2018. Currently, it is difficult to judge whether the stagnation of life expectancy in 2018 is temporary, and further analyses of life expectancy and contributing causes of death in the future are needed.

## Supporting information

**S1 Appendix.**
(XLSX)

**S1 Table. The population and number of deaths in Korea between 2003 and 2018.**
(DOCX)

**S2 Table. Broad cause-specific contributions to the change in life expectancy in four sub-periods (2003–2006, 2007–2010, 2011–2014, and 2015–2018) in Korea.**
(DOCX)

**S3 Table. Detailed cause-specific contributions (for 112 causes of deaths) to the change in life expectancy in 2003–2006, 2007–2010, 2011–2014, and 2015–2018 in Korea.** (DOCX)

## Acknowledgments

We thank the reviewers for their helpful comments on the interpretation of changes in life expectancy in Korea and the analysis method.

## Author Contributions

**Conceptualization:** Young-Ho Khang.

**Data curation:** Taejae Kim.

**Formal analysis:** Taejae Kim, Jinwook Bahk, Hwa Kyung Lim.

**Investigation:** Taejae Kim.

**Methodology:** Young-Ho Khang.

**Supervision:** Young-Ho Khang.

**Validation:** Young-Ho Khang.

**Writing – original draft:** Taejae Kim.

**Writing – review & editing:** Jinwook Bahk, Hwa Kyung Lim, Young-Ho Khang.

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
