## [Decision Letter · Decision Letter 0]

13 Aug 2020

PONE-D-20-16499

Stagnation of life expectancy in Korea in 2018: A cause-specific decomposition analysis

PLOS ONE

Dear Dr. Khang,

Thank you for submitting your manuscript to PLOS ONE. After careful consideration, we feel that it has merit but does not fully meet PLOS ONE’s publication criteria as it currently stands. Therefore, we invite you to submit a revised version of the manuscript that addresses the points raised during the review process.

We believe the paper is interesting and raises some important issues. But it is necessary major revision before it can be considered for publication. Please, see detailed comments by the reviewers in revising the paper. There are many issues to be considered:

1) paper needs better contextualization of the problem and contribution

2) better discussion of using only one year in the analysis. Please, see detailed discussion by reviewer.

3) Reviewers and myself are in doubt of some concepts using and methods. For instance, not very clear the procedure to calculate life expectancy, there are some issues on discussion the Arriaga method.

4) improve the discussion on the oldest-old mortality

5) better discussion of data quality

6) considered suggested references and approches proposed by reviewers. 

We look forward to receiving your revised manuscript.

Kind regards,

Bernardo Lanza Queiroz, Ph.D

Academic Editor

PLOS ONE

Journal Requirements:

Reviewers' comments:

Reviewer's Responses to Questions

**Comments to the Author**

1. Is the manuscript technically sound, and do the data support the conclusions?

Reviewer #1: Yes

Reviewer #2: Partly

2. Has the statistical analysis been performed appropriately and rigorously? 

Reviewer #1: Yes

Reviewer #2: Yes

3. Have the authors made all data underlying the findings in their manuscript fully available?

Reviewer #1: Yes

Reviewer #2: No

4. Is the manuscript presented in an intelligible fashion and written in standard English?

Reviewer #1: Yes

Reviewer #2: Yes

5. Review Comments to the Author

Reviewer #1: the answer to the main issue is simple. decrease um gains nas stagnation occurs because le is already high. 82.7 years. therefore the articule has to explicitally mention that and change the article in Light of this. your finfings indicate deaths occur only among the oldest old? you have that answer.

have to calculate contributions based also ok percent.

the discution is very long.

avoid using 0.0 or 0 in text

Reviewer #2: Please see the attachment for a version of this review including the figure.

According to the Human Mortality Database (www.mortality.org), which compounds 50 mostly rich countries and areas, since 1990 a yearly decrease in life expectancy at birth was observed in 17% of cases (16% for males and 21% for females), that is in 208 out of 1257 country-years . Does each of these 208 cases deserve a dedicated study? Certainly not. Does the Korean case stand apart as the only country in this list that has so far never experienced a drop in life expectancy? Probably yes. Should we tread carefully when studying a specific case of a specific year? Assuredly yes. Ignoring this general fact leads the authors to work with blinkers, by focusing on a single year while ignoring the general trends and the effect of variability, eventually having to squeeze every last drop of their results to the point of over-interpreting anecdotal evidences.

The aim of this paper is to investigate the causes of the stagnation of life expectancy in South Korea between 2017 and 2018. It is well conceived and written, the choice of data and methods is relevant, and the analyses are sound. However, the focus of this paper on the specificity of the year 2018 is justified by the fact that this year did not witness any improvement (or barely) in mortality for the first time since modern records. Still, if it can indeed be considered as an unprecedented event, it is part of a general pattern of decreasing rate of improvement in the last 15 years (Figure 1), and the fact that it happened during this particular year has probably as much to do with random fluctuations as with the underlying causes of this trend.

Figure 1: Yearly change in e0 in Korea (total population), HMD 2020

Figure 1 teaches us two things. First, given the overall slowdown in mortality improvements in the last 15 years, if we were to fit a linear trend to the yearly change in e0 (dotted line), we would expect life expectancy to come to a complete halt around the year 2029. Whether or not this really happens at this particular date, it means that such an eventuality has become more and more probable over time and is bound to happen within the next decade. The authors seem oblivious to this reality and repeatedly state that Korean life expectancy has been steadily increasing to the point of soon becoming the highest in the world (which is not necessarily incompatible with a slowing down of improvements, depending on what happens in other countries like Japan). Considering this trend makes the absence of progress in 2018 much less sudden and unexpected than the authors present it when they write for instance that “The unprecedented increase in life expectancy in Korea suddenly came to a halt in 2018.” (line 204).

Secondly, progresses in life expectancy do not come without some natural yearly variability. As shown in Figure 1, the observations are not following that closely the trend line. Between 2003 and 2018, while the mean of annual gains in life expectancy was 0.371, their standard deviation was 0.158, meaning that about half of the yearly change is driven by conjunctural circumstances, i.e. a mix of punctual events like flue epidemics or heat waves, and natural stochasticity (the other half being driven by the causes of the downward trend). For instance, between 2008 and 2013, annual fluctuations in e0 were +0.44, +0.17, +0.44, +0.17, and +0.52, leading to variations (ca. ±0.3) almost as large as the mean annual change across 2003-2018. It is thus misleading to state that “from 2000 to 2017, the rate of increase was consistent” (lines 134-135). Given the level of change in the previous years, it is thus likely that at least part of the drop registered in 2018 was due to conjunctural fluctuations that have little meaning in the general stagnation of life expectancy in Korea.

These important general remarks lead to the following specific necessary improvements in the manuscript.

1. The authors need to put the specific year 2018 in perspective with the general downward (not stable) trend that preceded it. This applies specifically to the introduction, notably by avoiding to present 2018 as an extraordinary year when mortality suddenly stopped improving out of the blue.

2. The authors need to refrain from making comparisons with other cases such as the American and British ones (e.g. lines 66 to 75), which bear no resemblance to the one of Korea (as noted by themselves on lines 198 to 203). It would be more interesting for instance to draw comparisons with other East-Asian countries with high life expectancy.

3. More attention needs to be paid to the dimensions of variability and uncertainty in the data and the results. I suggest for instance considering resampling methods to compute confidence intervals for the decomposition (Bergeron-Boucher et al., 2019), which would give an indication of the importance of stochasticity in the results. Another option could be to divide the period into several sub-periods (e.g. 2000-2005, 2006-2009, 2010-2014 and 2015-2018), which would (1) allow to see the progressive evolution in the mortality patterns rather than opposing the single year 2018 to the mean of all previous years, (2) work with samples of similar size and thus submitted to a similar level of variability, and (3) possibly avoid having to rely on dubious explanations like copycat suicides (Why would this behavior suddenly increase in this particular year? Is there evidence that the suicide of a specific well-known figure was directly followed by a strong increase in the public?), which feels at the moment like an ad-hoc explanation for a conjunctural event.

4. Why not considering harvesting effects in your discussion? The years 2016 and 2017 both registered annual improvements above what was expectable from the general trend (Figure 1). Could the subsequent excess mortality in 2018 be partly due to “overdue” deaths? A similar effect could for instance explain the drop in 2015.

Additionally, I have two requests regarding the data.

1. I would like the authors to be more specific in their appreciation of the data quality, especially for the period 2000-2002. In their documentation, the Human Mortality Database indicate a historical strong underregistration of infant mortality (also mentioned by the authors on lines 97-104). For this reason, the HMD chose to start their series in 2003, not in 2000. Could you explain the reason behind this difference with your study? How do you value the quality of your data between 2000 and 2002, have you explored the trends of mortality before the first birthday, and do you think this could affect your results?

2. What is the proportion of ill-defined causes of death (R00-R99) in the dataset, and how is this proportion evolving over time? Based on another publication, you suggest that strokes might be responsible for a large part of ill-defined deaths (line 314), although this claim is not substantiated in your study, nor the one you cite. Because they are notoriously hard to interpret and their evolution might reflect changes either in etiology or in registration (or coding), ill-defined deaths are typically proportionally redistributed across all other causes of death, unless it is possible to establish a correlation with specific causes of death across time or subnational entities (e.g. Meslé & Vallin, 2012). I suggest that the authors adopt one of these techniques in order to eliminate ill-defined causes of death from the analysis.

Finally, I would like the authors to clarify what they mean in the following two paragaphs of the methods section.

1. I do not understand what they mean by “Life expectancy at birth is calculated by dividing the number of surviving 0-year-olds by the stationary population of 0-year-olds. This study calculated the stationary population of each age based on the surviving number of individuals at each age, the total stationary population beyond a certain age, and the life expectancy at birth for each age category.” (lines 111-114). Computing life expectancy does not require working with stationary populations, so I do not understand what was the aim of this approach, nor how it was practically implemented. Was it designed to correct for the under-registration of infant deaths?

2. It is not my understanding that the Arriaga method distinguishes between direct, indirect and interaction effects (lines 119-123). I belive that it only gives one contribution by age group, which can be further decomposed by cause of death using the assumption of proportionality between age-specific death rates and age-specific contributions. I would like the authors to check this point and either use these three effects in their results, or drop this reference that only introduces confusion since only one effect is later presented in the results.

References

Bergeron-Boucher, M.-P., Oeppen, J., Holm, N. V., Nielsen, H. M., Lindahl-Jacobsen, R., & Wensink, M. J. (2019). Understanding differences in cancer survival between populations: A new approach and application to breast cancer survival differentials between Danish regions. International journal of environmental research and public health, 16(17), 3093.

Meslé, F., & Vallin, J. (2012). General Trends in Mortality by Cause. In Mortality and Causes of Death in 20th-Century Ukraine (pp. 153-172): Springer.

6. PLOS authors have the option to publish the peer review history of their article (what does this mean?). If published, this will include your full peer review and any attached files.

Reviewer #1: **Yes: **Carla Jorge Machado

Reviewer #2: **Yes: **A. Remund

---

## [Author Response · Author response to Decision Letter 0]

21 Oct 2020

Authors’ Responses to Comments

We thank reviewers for their helpful comments and suggestions. Based on the reviewer’s comments (especially reviewer #2’s comments), we reanalyzed the data and extensively revised our main text. We have responded to each comment made by both reviewers below. 

Editorial requests: 

We believe the paper is interesting and raises some important issues. But it is necessary major revision before it can be considered for publication. Please, see detailed comments by the reviewers in revising the paper. There are many issues to be considered:

1) paper needs better contextualization of the problem and contribution

(OUR RESPONSE) Based on the comments from the reviewers, we deleted many sentences in the Introduction and Discussion sections and newly added Fig. 2 and Tables 1 and 2 based on the reviewer #2’s comment.

2) better discussion of using only one year in the analysis. Please, see detailed discussion by reviewer.

(OUR RESPONSE) The reviewer #2 indicated any potential problem using only one year in our analysis and suggested an option to divide the study period into several sub-periods. We conducted such analyses using four sub-periods and presented findings in the revised version. We newly added Tables 1 and 2 and associated sentences in the Result and Discussion sections. Please see our revised version.

3) Reviewers and myself are in doubt of some concepts using and methods. For instance, not very clear the procedure to calculate life expectancy, there are some issues on discussion the Arriaga method.

(OUR RESPONSE) Regarding the calculation of the life expectancy and the Arriaga’s method, we clarified the methods in our revised Method section. Please see lines 127-142 in the revised version with track changes.

4) improve the discussion on the oldest-old mortality

(OUR RESPONSE) We used data for oldest old age groups (i.e., 80+) as prior paper by us indicated that the mortality rate among 80+ age groups were important (lines 106-111 in the revised version with track changes). Regarding the mortality rates among older age groups, we inserted that “These findings imply that the stagnating life expectancy in 2018 might have resulted from a decreasing rate of mortality reduction in causes of death in older age groups.” Please see lines 277-278 in the revised version with track changes.

5) better discussion of data quality

(OUR RESPONSE) The reviewer #2 indicated any potential issue using 2000-2002 mortality data in S Korea. Based on this comment and the comment documented in the Human Mortality Database (cited as the reference #9), we decided to use data for the period between 2003 and 2018 rather than 2000-2018 (see lines 118-119 in the revised version with track changes). Subsequently, all the Figures and Tables in our paper have been revised.

6) considered suggested references and approches proposed by reviewers. 

(OUR RESPONSE) We reviewed the reference on resampling methods for confidence intervals of decomposition analysis suggested by the Reviewer #2 and cited the reference (Bergeron-Boucher et al., 2019). Regarding the approach for analyzing the data, the reviewer #2 suggested two options. We chose the second option to divide the study period into several sub-periods. Subsequently, we newly added Tables 1 and 2 and extensively revised our Result and Discussion sections. Basically, we rewrote our discussion.

Reviewer(s)' Comments to Author: 

Reviewer: 1 

(1) the answer to the main issue is simple. decrease um gains nas stagnation occurs because le is already high. 82.7 years. therefore the articule has to explicitally mention that and change the article in Light of this. 

(OUR RESPONSE) This study is to explore the reason why life expectancy in S Korea has stagnated in 2018. We do not believe that the answer is because life expectancy is already high. 

(2) your finfings indicate deaths occur only among the oldest old? you have that answer.

(OUR RESPONSE) We used data for oldest old age groups (i.e., 80+) as prior paper by us indicated that the mortality rate among 80+ age groups were important (lines 106-111 in the revised version with track changes). Regarding the mortality rates among older age groups, we inserted that “These findings imply that the stagnating life expectancy in 2018 might have resulted from a decreasing rate of mortality reduction in causes of death in older age groups.” Please see lines 277-278 in the revised version with track changes.

(3) have to calculate contributions based also ok percent.

(OUR RESPONSE) Based on the reviewer’s comment, we added % in our Tables 1 and 2.

(4) the discution is very long.

(OUR RESPONSE) We deleted discussion points related with stagnated life expectancies in the UK and US. Based on our new analysis results, we rewrote and shortened our discussion. Please see our new Discussion section.

(5) avoid using 0.0 or 0 in text

(OUR RESPONSE) We do not believe that 0.0 cannot be used in our paper. For clarity, we retained the numbers in our revised version.

Reviewer: 2 

According to the Human Mortality Database (www.mortality.org), which compounds 50 mostly rich countries and areas, since 1990 a yearly decrease in life expectancy at birth was observed in 17% of cases (16% for males and 21% for females), that is in 208 out of 1257 country-years1. Does each of these 208 cases deserve a dedicated study? Certainly not. Does the Korean case stand apart as the only country in this list that has so far never experienced a drop in life expectancy? Probably yes. Should we tread carefully when studying a specific case of a specific year? Assuredly yes. Ignoring this general fact leads the authors to work with blinkers, by focusing on a single year while ignoring the general trends and the effect of variability, eventually having to squeeze every last drop of their results to the point of over-interpreting anecdotal evidences.

The aim of this paper is to investigate the causes of the stagnation of life expectancy in South Korea between 2017 and 2018. It is well conceived and written, the choice of data and methods is relevant, and the analyses are sound. However, the focus of this paper on the specificity of the year 2018 is justified by the fact that this year did not witness any improvement (or barely) in mortality for the first time since modern records. Still, if it can indeed be considered as an unprecedented event, it is part of a general pattern of decreasing rate of improvement in the last 15 years (Figure 1), and the fact that it happened during this particular year has probably as much to do with random fluctuations as with the underlying causes of this trend.

Figure 1 teaches us two things. First, given the overall slowdown in mortality improvements in the last 15 years, if we were to fit a linear trend to the yearly change in e0 (dotted line), we would expect life expectancy to come to a complete halt around the year 2029. Whether or not this really happens at this particular date, it means that such an eventuality has become more and more probable over time and is bound to happen within the next decade. The authors seem oblivious to this reality and repeatedly state that Korean life expectancy has been steadily increasing to the point of soon becoming the highest in the world (which is not necessarily incompatible with a slowing down of improvements, depending on what happens in other countries like Japan). Considering this trend makes the absence of progress in 2018 much less sudden and unexpected than the authors present it when they write for instance that “The unprecedented increase in life expectancy in Korea suddenly came to a halt in 2018.” (line 204).

Secondly, progresses in life expectancy do not come without some natural yearly variability. As shown in Figure 1, the observations are not following that closely the trend line. Between 2003 and 2018, while the mean of annual gains in life expectancy was 0.371, their standard deviation was 0.158, meaning that about half of the yearly change is driven by conjunctural circumstances, i.e. a mix of punctual events like flue epidemics or heat waves, and natural stochasticity (the other half being driven by the causes of the downward trend). For instance, between 2008 and 2013, annual fluctuations in e0 were +0.44, +0.17, +0.44, +0.17, and +0.52, leading to variations (ca. ±0.3) almost as large as the mean annual change across 2003-2018.

(OUR RESPONSE) Many thanks for your insightful comments helpful for our revision. Based on your comments, we added Fig. 2 and extensively rewrote our discussion. We think that there were several points needing our response and amendment.

First, you started your comment with data from the Human Mortality Database on yearly life expectancy changes in 50 rich countries and areas. It is true that we could see yearly decrease or stagnation in life expectancy for several years in these countries, but our points here would be that (1) many of those countries have a relatively smaller population than South Korea (50 millions) and thus may have a greater likelihood of random variability in yearly changes in life expectancy and (2) the speed of yearly life expectancy increase in South Korea has been relatively greater in the past decades than most of the rich countries and areas. 

Second, you assessed time trends in yearly life expectancy changes in S Korea between 2003 and 2018 and presented Fig. 1 in your comment above showing a decreasing trend in yearly life expectancy changes. As you can see in our Response Letter Figure 1 below, the beta coefficient (slope) of yearly changes in life expectancy per calendar year between 2003 and 2018 was -0.0211. However, it would be more reasonable to see the background trends in yearly life expectancy changes per calendar year between 2003 and 2017 (rather than 2018) (Response Letter Figure 2). The beta (slope) for this was -0.015. 

Response Letter Figure 1. Yearly life expectancy changes between 2003 and 2018.

Response Letter Figure 2. Yearly life expectancy changes between 2003 and 2017.

Regarding this point on yearly life expectancy changes, we inserted Fig. 2 and associated sentences in the Result section (see 165-168 in the revised version with track changes). In addition, we also tried to avoid assertive expressions over the paper (in the Discussion section) regarding the life expectancy in 2018 and to accept any possibility of random variation in life expectancy. Please see our revised version.

Third, since you indicated in your comments that the life expectancy in 2018 might be considered as random variation, we tried to examine whether the pattern in 2018 is a statistically significant decrease. The reason that the magnitude of increase in life expectancy in 2018 decreased greatly would be because the life expectancy in 2018 was estimated to be much lower than the trend of life expectancy by year. We performed an outlier test for the life expectancy in 2018 based on the regression analysis on life expectancy by year. The outlierTest () function in the car package in R was used, and the result of the outlier test is as follows:

No Studentized residuals with Bonferroni p < 0.05

Largest |rstudent|: rstudent unadjusted p-value Bonferroni p

16 -2.698004 0.018263 0.29221

The studentized residual for the year 2018 was the largest. However, since the Bonferroni P-value (=0.29221) for the null hypothesis (H0: life expectancy in 2018 is not an outlier) was greater than the significance level of 0.05, H0 cannot be rejected, that is, the life expectancy in 2018 was not an outlier. Therefore, we could say that the life expectancy in 2018 is lower than the trend of life expectancy by year, but it is not an outlier. Regarding this point, we added several sentences in the revised version (lines 168-173).

Fourth, the reviewer documented in the review comment that we repeatedly state that Korean life expectancy has been steadily increasing to the point of soon becoming the highest in the world. This point is not our assertion but based on the previous Lancet paper led by Majid Ezzati in Imperial College London (Kontis et al., 2017). Considering the comment, we deleted the sentence in the Introduction section (lines 63-68 in the revised version with track changes).

It is thus misleading to state that “from 2000 to 2017, the rate of increase was consistent” (lines 134-135). Given the level of change in the previous years, it is thus likely that at least part of the drop registered in 2018 was due to conjunctural fluctuations that have little meaning in the general stagnation of life expectancy in Korea.

(OUR RESPONSE) Considering the point the reviewer indicated, we deleted ‘consistent’ and changed the sentence as follows: “From 2003 to 2017, the annual increase in life expectancy ranged between 0.144 years and 0.579 years, and the mean annual rate of increase was 0.398 years (0.426 years for males and 0.361 for females)”. Please see lines 153-157 in the revised version with track changes.

These important general remarks lead to the following specific necessary improvements in the manuscript. 

1. The authors need to put the specific year 2018 in perspective with the general downward (not stable) trend that preceded it. This applies specifically to the introduction, notably by avoiding to present 2018 as an extraordinary year when mortality suddenly stopped improving out of the blue.

(OUR RESPONSE) In the Introduction section, we added a sentence that the magnitude of annual increase in life expectancy has varied (lines 61-62 in the revised version with track changes). In addition, we also deleted the sentence in the discussion section describing that 2018 is an extraordinary year (e.g., The unprecedented increase in life expectancy in Korea suddenly came to a halt in 2018.). Please see the rewritten Discussion section.

2. The authors need to refrain from making comparisons with other cases such as the American and British ones (e.g. lines 66 to 75), which bear no resemblance to the one of Korea (as noted by themselves on lines 198 to 203). It would be more interesting for instance to draw comparisons with other East-Asian countries with high life expectancy. 

(OUR RESPONSE) Based on the comment, we deleted the sentences in the Introduction and Discussion sections and comparing the Korean case with the life expectancy stagnation in the US and the UK (lines 77-88 and 311-321 in the revised version with track changes).

3. More attention needs to be paid to the dimensions of variability and uncertainty in the data and the results. I suggest for instance considering resampling methods to compute confidence intervals for the decomposition (Bergeron-Boucher et al., 2019), which would give an indication of the importance of stochasticity in the results. Another option could be to divide the period into several sub-periods (e.g. 2000-2005, 2006-2009, 2010-2014 and 2015-2018), which would (1) allow to see the progressive evolution in the mortality patterns rather than opposing the single year 2018 to the mean of all previous years, (2) work with samples of similar size and thus submitted to a similar level of variability, and (3) possibly avoid having to rely on dubious explanations like copycat suicides (Why would this behavior suddenly increase in this particular year? Is there evidence that the suicide of a specific well-known figure was directly followed by a strong increase in the public?), which feels at the moment like an ad-hoc explanation for a conjunctural event.

(OUR RESPONSE) We have several points to respond. 

First, the reviewer suggested resampling methods to compute confidence intervals for the decomposition. In this study, we used aggregate population and mortality data grouped by calendar year, sex, and age group (and causes of death for mortality data). This point has been clarified in the revised version (lines 98-99 and 426-427). As indicated in the paper the reviewer suggested (Bergeron-Boucher et al., 2019), “confidence intervals for the background effects (age and survival) using the above described method cannot be estimated, as these effects are based on aggregated data only (Meslé & Vallin, 2012).” Thus, we could not present confidence intervals for the decomposition by resampling methods. Regarding this point, we added a study limitation in the revised version (lines 426-428).

Second, the reviewer also suggested another option to divide the period into several sub-periods. Based on this suggestion, we conducted decomposition analyses after dividing the period in to four sub-periods (2003-2006, 2007-2010, 2011-2014, and 2015-2018) and added Tables 1 and 2, and S3 Table. Subsequently, we added associated sentences in the Result section and rewrote our discussion. Please see our revised version.

Third, since our new analyses to divide the period in to four sub-periods produced any meaningful effect by suicide on life expectancy changes, we deleted discussion points on suicide.

4. Why not considering harvesting effects in your discussion? The years 2016 and 2017 both registered annual improvements above what was expectable from the general trend (Figure 1). Could the subsequent excess mortality in 2018 be partly due to “overdue” deaths? A similar effect could for instance explain the drop in 2015.

(OUR RESPONSE) As we responded to the previous comment, we deleted discussion points on suicide as our new analyses to divide the period in to four sub-periods produced any meaningful effect by suicide on life expectancy changes, 

Additionally, I have two requests regarding the data.

1. I would like the authors to be more specific in their appreciation of the data quality, especially for the period 2000-2002. In their documentation, the Human Mortality Database indicate a historical strong underregistration of infant mortality (also mentioned by the authors on lines 97-104). For this reason, the HMD chose to start their series in 2003, not in 2000. Could you explain the reason behind this difference with your study? How do you value the quality of your data between 2000 and 2002, have you explored the trends of mortality before the first birthday, and do you think this could affect your results?

(OUR RESPONSE) In our original version, we chose to start the time series in 2000 since the KOSIS (the official data sharing website for Statistics Korea) provides stratified age groups of 80-84, 85-89, and 90 and above since 2000 and Statistics Korea started to include data from crematorium for infant mortality. However, based on the reviewer’s point, we decided to use data for the period 2003-2018 rather than the period 2000-2018. Regarding this change, we added the point in the revised Method section (lines 118-119). Subsequently, all the Figures and Tables and associated sentences have been revised.

2. What is the proportion of ill-defined causes of death (R00-R99) in the dataset, and how is this proportion evolving over time? Based on another publication, you suggest that strokes might be responsible for a large part of ill-defined deaths (line 314), although this claim is not substantiated in your study, nor the one you cite. Because they are notoriously hard to interpret and their evolution might reflect changes either in etiology or in registration (or coding), ill-defined deaths are typically proportionally redistributed across all other causes of death, unless it is possible to establish a correlation with specific causes of death across time or subnational entities (e.g. Meslé & Vallin, 2012). I suggest that the authors adopt one of these techniques in order to eliminate ill-defined causes of death from the analysis. 

(OUR RESPONSE) The proportion of ill-defined causes of death (R00-R99) is presented below and is relatively stable. Regarding the reviewer’s point on strokes which is not substantiated in our paper, we deleted the sentence in our revised version. In this study, we did not redistribute ill-defined causes but retained the category in our result because we thought that proportional redistribution of ill-defined causes might only proportionally increase the contribution of each cause of death to life expectancy increase. In our study limitation, we added a point on ill-defined causes (lines 431-433).

Response Letter Figure 4. Proportion of ill-defined causes (R00-R99) among total death in Korea between 2003 and 2018.

Finally, I would like the authors to clarify what they mean in the following two paragaphs of the methods section.

1. I do not understand what they mean by “Life expectancy at birth is calculated by dividing the number of surviving 0-year-olds by the stationary population of 0-year-olds. This study calculated the stationary population of each age based on the surviving number of individuals at each age, the total stationary population beyond a certain age, and the life expectancy at birth for each age category.” (lines 111-114). Computing life expectancy does not require working with stationary populations, so I do not understand what was the aim of this approach, nor how it was practically implemented. Was it designed to correct for the under-registration of infant deaths?

(OUR RESPONSE) Based on this comment, we deleted the previous sentences (lines 128-132 in the revised version with track changes) and revised sentence as follows: We constructed annual abridged life tables with 5-year age groups and calculated life expectancy at birth using life table method. Please see lines 127-128 in the revised version.

2. It is not my understanding that the Arriaga method distinguishes between direct, indirect and interaction effects (lines 119-123). I belive that it only gives one contribution by age group, which can be further decomposed by cause of death using the assumption of proportionality between age-specific death rates and age-specific contributions. I would like the authors to check this point and either use these three effects in their results, or drop this reference that only introduces confusion since only one effect is later presented in the results. 

(OUR RESPONSE) You are right. Based on this comment, we removed the sentences (lines 138-142 in the revised version with track changes). 

References

Bergeron-Boucher, M.-P., Oeppen, J., Holm, N. V., Nielsen, H. M., Lindahl-Jacobsen, R., & Wensink, M. J. (2019). Understanding differences in cancer survival between populations: A new approach and application to breast cancer survival differentials between Danish regions. International journal of environmental research and public health, 16(17), 3093.

Meslé, F., & Vallin, J. (2012). General Trends in Mortality by Cause. In Mortality and Causes of Death in 20th-Century Ukraine (pp. 153-172): Springer.

---

## [Decision Letter · Decision Letter 1]

2 Dec 2020

PONE-D-20-16499R1

Stagnation of life expectancy in Korea in 2018: A cause-specific decomposition analysis

PLOS ONE

Dear Dr. Khang,

Thank you for submitting your manuscript to PLOS ONE. After careful consideration, we feel that it has merit but does not fully meet PLOS ONE’s publication criteria as it currently stands. Therefore, we invite you to submit a revised version of the manuscript that addresses the points raised during the review process.

there are a very few points in editing to be adjusted - please review thatI agree that data in the paper should be available - Figshare is an option

We look forward to receiving your revised manuscript.

Kind regards,

Bernardo Lanza Queiroz, Ph.D

Academic Editor

PLOS ONE

Reviewers' comments:

Reviewer's Responses to Questions

**Comments to the Author**

1. If the authors have adequately addressed your comments raised in a previous round of review and you feel that this manuscript is now acceptable for publication, you may indicate that here to bypass the “Comments to the Author” section, enter your conflict of interest statement in the “Confidential to Editor” section, and submit your "Accept" recommendation.

Reviewer #1: (No Response)

Reviewer #2: All comments have been addressed

2. Is the manuscript technically sound, and do the data support the conclusions?

Reviewer #1: Yes

Reviewer #2: Yes

3. Has the statistical analysis been performed appropriately and rigorously? 

Reviewer #1: Yes

Reviewer #2: Yes

4. Have the authors made all data underlying the findings in their manuscript fully available?

Reviewer #1: No

Reviewer #2: No

5. Is the manuscript presented in an intelligible fashion and written in standard English?

Reviewer #1: (No Response)

Reviewer #2: Yes

6. Review Comments to the Author

Reviewer #1: Please change the 0.0 or 0 in your paper. It is better to read there is a null increase or there no decrease/increase than reading 0.0. Change that.

You provide no explanation in the text for not writing down that the expectation is already high. Do that. I have a question, I am the reviewer. The reader needs to have all the information and this is a peer review journal. You cannot simply say I don't believe that. It is not the proper way to answer. Provide reasoning for your choices.

Reviewer #2: All necessary changes have been made to the manuscript. I am fully satisfied by the modifications applied by the authors, and would like to thank them for their detailed and clear responses. In the new version, some tables and figures are partly overlapping, and it might be a good idea to move some of them (for instance the tables) to the appendix. This is a matter to see with the editorial team though. For the record, resampling methods are also possible on aggregate data, using a Monte Carlo approach based on a theoretical distribution of the aggregate data (in this case Poisson), but the authors' choice of avoiding that path is understandable given that the division of the observation into several sub-periods was sucessful in clarifying the conclusions. Regarding the availability of the data, I let the editor judge, but I assume that providing the age- and cause-specific death rates for the different subperiods would be enough (e.g. as printed tables or a csv file).

7. PLOS authors have the option to publish the peer review history of their article (what does this mean?). If published, this will include your full peer review and any attached files.

Reviewer #1: No

Reviewer #2: **Yes: **Adrien Remund

---

## [Author Response · Author response to Decision Letter 1]

6 Dec 2020

Authors’ Responses to Comments

Based on the editor’s and reviewers’ comments, we additionally revised our manuscript. All changes were highlighted in red on the revised manuscript.

Editorial requests: 

1) there are a very few points in editing to be adjusted - please review that

(OUR RESPONSE) Based on the reviewers’ comment, we revised some expressions in our paper and moved a Table to the appendix with deleting an overlapping Table. Please see our revised version.

2) I agree that data in the paper should be available - Figshare is an option

(OUR RESPONSE) All data used in this study can be accessed via KOSIS (https://kosis.kr/eng/index/index.do) - Population Statistics Based on Resident Registration, Deaths and Death rates by cause (236 item), sex, and age. However, based on the reviewer’s and editor’s suggestion, we submitted the data for the number of deaths and population released by Statistics Korea.

Reviewer(s)' Comments to Author: 

Reviewer: 1 

Please change the 0.0 or 0 in your paper. It is better to read there is a null increase or there no decrease/increase than reading 0.0. Change that.

(OUR RESPONSE) We changed the expression ‘an increase of 0.0 years’ to ‘a null increase’. Please see lines 51 in the revised version with track changes.

You provide no explanation in the text for not writing down that the expectation is already high. Do that. I have a question, I am the reviewer. The reader needs to have all the information and this is a peer review journal. You cannot simply say I don't believe that. It is not the proper way to answer. Provide reasoning for your choices.

(OUR RESPONSE) Based on this comment, we inserted two sentences in the Introduction section (lines 61-64 in the revised version with track changes). “One possible explanation for the stagnation would be that Korea’s life expectancy in 2017 is already high (82.7 years) and has reached its limit. Age- and cause-specific contributions to the life expectancy increase may also provide an answer to such explanations.” We found that the contribution of cardiovascular disease (I00-I99) to life expectancy increase is gradually decreasing over the period 2003-2018, which means that mortality from cerebrovascular disease is reaching its limit. We think that this point would be partly related with the explanation that Korea’s life expectancy is already high. Thus, we also inserted a sentence in the discussion section, “partly supporting an explanation that Korea’s life expectancy is already high” (lines 245-246 in the revised version with track changes).

Reviewer: 2 

All necessary changes have been made to the manuscript. I am fully satisfied by the modifications applied by the authors, and would like to thank them for their detailed and clear responses. 

(OUR RESPONSE) Thank you.

In the new version, some tables and figures are partly overlapping, and it might be a good idea to move some of them (for instance the tables) to the appendix. This is a matter to see with the editorial team though.

(OUR RESPONSE) Since the contents of Table S2 and Table 2 overlapped, Table S2 was deleted and Table 2 was moved to the appendix as Table S2. 

For the record, resampling methods are also possible on aggregate data, using a Monte Carlo approach based on a theoretical distribution of the aggregate data (in this case Poisson), but the authors' choice of avoiding that path is understandable given that the division of the observation into several sub-periods was sucessful in clarifying the conclusions.

(OUR RESPONSE) Thank you for this comment. Based on the comment, we modified our study limitation section. Please see lines 273-275 in the revised version with track changes.

Regarding the availability of the data, I let the editor judge, but I assume that providing the age- and cause-specific death rates for the different subperiods would be enough (e.g. as printed tables or a csv file)

(OUR RESPONSE) All data used in this study were provided by KOSIS (https://kosis.kr/eng/index/index.do) - Population Statistics Based on Resident Registration, Deaths and Death rates by cause (236 item), sex, and age. You can access the data directly from KOSIS, but for your convenience, we submitted the data in xlsx file format.

---

## [Editor Report · Decision Letter 2]

9 Dec 2020

Stagnation of life expectancy in Korea in 2018: A cause-specific decomposition analysis

PONE-D-20-16499R2

Dear Dr. Khang,

We’re pleased to inform you that your manuscript has been judged scientifically suitable for publication and will be formally accepted for publication once it meets all outstanding technical requirements.

Kind regards,

Bernardo Lanza Queiroz, Ph.D

Academic Editor

PLOS ONE

Additional Editor Comments (optional):

Thank you for considering and working on all comments and suggestions. This is a very interesting paper. 
---

## [Editor Report · Acceptance letter]

11 Dec 2020

PONE-D-20-16499R2 

Stagnation of life expectancy in Korea in 2018: A cause-specific decomposition analysis 

Dear Dr. Khang:

I'm pleased to inform you that your manuscript has been deemed suitable for publication in PLOS ONE. Congratulations! Your manuscript is now with our production department. 

Kind regards, 

on behalf of

Dr. Bernardo Lanza Queiroz 

Academic Editor

PLOS ONE